# Cognitive Decline in Early and Premature Menopause

**DOI:** 10.3390/ijms24076566

**Published:** 2023-03-31

**Authors:** Marta Sochocka, Julia Karska, Magdalena Pszczołowska, Michał Ochnik, Michał Fułek, Katarzyna Fułek, Donata Kurpas, Justyna Chojdak-Łukasiewicz, Anna Rosner-Tenerowicz, Jerzy Leszek

**Affiliations:** 1Hirszfeld Institute of Immunology and Experimental Therapy, Polish Academy of Sciences, 53-114 Wroclaw, Poland; 2Department of Psychiatry, Wroclaw Medical University, 50-367 Wroclaw, Poland; 3Department of Internal Medicine, Occupational Diseases, Hypertension and Clinical Oncology, Wroclaw Medical University, 50-556 Wroclaw, Poland; 4Department and Clinic of Otolaryngology, Head and Neck Surgery, Wroclaw Medical University, 50-556 Wroclaw, Poland; 5Department of Family Medicine, Wroclaw Medical University, 51-141 Wroclaw, Poland; 6Department of Neurology, Wroclaw Medical University, 50-566 Wroclaw, Poland; 72nd Department of Gynecology and Obstetrics, Wroclaw Medical University, 50-556 Wroclaw, Poland

**Keywords:** menopause, mild cognitive impairment, Alzheimer’s disease, depression, dementia, estrogen therapy

## Abstract

Early and premature menopause, or premature ovarian insufficiency (POI), affects 1% of women under the age of 40 years. This paper reviews the main aspects of early and premature menopause and their impact on cognitive decline. Based on the literature, cognitive complaints are more common near menopause: a phase marked by a decrease in hormone levels, especially estrogen. A premature reduction in estrogen puts women at a higher risk for cardiovascular disease, parkinsonism, depression, osteoporosis, hypertension, weight gain, midlife diabetes, as well as cognitive disorders and dementia, such as Alzheimer’s disease (AD). Experimental and epidemiological studies suggest that female sex hormones have long-lasting neuroprotective and anti-aging properties. Estrogens seem to prevent cognitive disorders arising from a cholinergic deficit in women and female animals in middle age premature menopause that affects the central nervous system (CNS) directly and indirectly, both transiently and in the long term, leads to cognitive impairment or even dementia, mainly due to the decrease in estrogen levels and comorbidity with cardiovascular risk factors, autoimmune diseases, and aging. Menopausal hormone therapy from menopause to the age of 60 years may provide a “window of opportunity” to reduce the risk of mild cognitive impairment (MCI) and AD in later life. Women with earlier menopause should be taken care of by various specialists such as gynecologists, endocrinologists, neurologists, and psychiatrists in order to maintain their mental health at the highest possible level.

## 1. Introduction

Menopause is defined by the World Health Organization (WHO) as the permanent cessation of menstrual periods due to the loss of ovarian activity without any other pathological or physiological cause. Menopause is recognized after a woman has experienced 12 months of amenorrhea. Menopause usually occurs at around age 50, but in some cases can be triggered between the ages of 41 and 45 years (early menopause) or even at age 40 (premature menopause). Premature menopause differs from menopause which occurs at around the average age (45–55 years). The occurrence of premature menopause means that the ovaries are not working properly and have stopped producing eggs years before they normally would [1]. Early and premature menopause may occur naturally or may be caused by chemotherapy or surgical interventions such as bilateral oophorectomy (ovariectomy). Premature menopause is estimated to affect 1% of women under the age of 40 years. Women with premature menopause are at risk of premature death, osteoporosis, ischemic heart disease, and infertility [2]. The term premature menopause, primary ovarian insufficiency (POI), and premature ovarian failure (POF) are often used interchangeably. POF is mostly an idiopathic state, but sometimes it may be caused by autoimmune disorders, genetic causes, infections or inflammatory conditions, enzyme deficiencies, or metabolic syndromes [1]. 

## 2. Methodology

In this narrative review, we focused on obtaining a broad perspective on the topic of the role of the undertime menopause period and mental health in humans. We hypothesized that early and premature menopause are associated with cognitive disabilities and dementia. The authors undertook a structured search of bibliographic databases, such as PubMed, Google Scholar, and Web of Science, for peer-reviewed research in the literature using a hypothesis and inclusion/exclusion criteria. The most relevant and up-to-date research was included. The primary search strategy was conducted using the following keywords: “premature menopause” OR “early menopause” OR “ovarian insufficiency” OR ,,dementia”, “Alzheimer’s Disease” AND “cognitive decline”. The inclusion criteria for selection were articles written in English published between 2000 and 2023, clinical studies, in vivo and in vitro studies, systematic reviews, and meta-analyses. Three authors decided which studies to include for review by consensus, with 111 resources selected for inclusion which was primary found in 151 positions. Exclusion criteria included any articles that failed to involve items described in the inclusion criteria or those that repeated a combination of phrases.

## 3. The Causes of Menopause

It is widely known that the ovarian reserve is established during prenatal life and peaks at 18–22 weeks post-conception when the number of oocytes reaches 6–7 million. A woman is born with a number of follicles, which, after the end of puberty, range from 300,000 to 400,000 in both ovaries. The ovaries also produce the hormones estradiol and progesterone, which regulate menstruation and ovulation [3]. Menopause is defined by the absence of a menstrual period for at least 12 months, after which women are no longer capable of getting pregnant. Menopause is related to the ovaries running out of suitable follicles from which ova are produced. Menopause typically occurs between the ages of 49 and 52 years and marks the transition from the reproductive to the non-reproductive period of life [4]. For some women, however, this process begins earlier than expected. Menopause comes prematurely to 1/250 women under 30 and 1/100 women under 40 [5]. Premature or early menopause can be spontaneous or induced. The majority of women who have expressed premature ovarian insufficiency (POI) have had normal puberty with regular cycles [6]. Different situations are the following: induced menopause under radiotherapy, chemotherapy, or the surgical removal of the ovaries and POI where menstruation is suddenly and permanently discontinued [7,8,9].

Recently the role of gut microbiomes in the changes in body composition and metabolic risk factors experienced by menopausal women have been considered. Although the gut microbiome is a key regulator of metabolism and influences the metabolism of estrogens, this mechanism remains largely unexplored. It was shown that in post-ovariectomy rodents, gut microbiome alterations were observed [10]. They were also associated with increased adiposity, decreased metabolic rate, insulin resistance, and changes attenuated by estrogen administration. Future research is needed to establish a causal link: the interrelationship between menopause and the gut microbiome [10]. The human gut microbiome has a modifiable nature and changes with aging. It displays sex differences, suggesting the influence of sex hormones. Low levels of estradiol and progesterone may lead to the permeability of the gut barrier and thus allow microbial translocation [11]. Research has shown that menopause and/or low estrogen were linked with reduced gut microbiome diversity and a greater similarity to men in microbiome composition [12].

## 4. The Causes and Risk Factors of Premature and Early Menopause

The causes that lead to premature menopause have not been precisely determined. Genetic factors probably play an important role since a positive family history is present in 50% of women with premature menopause [13]. Two functioning X chromosomes are needed for normal ovarian function. Genetic conditions which affect X chromosomes are related to premature menopause [14]. The possible causes of early and premature menopause are listed in Table 1 and Table 2. 

There are some factors that may contribute to follicular depletion and that are associated with the early onset of menopause. The most important of these is family history, with the risk increasing up to 12 times. It is suggested that heterozygous deleterious variants of *BUB1B* are involved in late-onset POI and related disorders [24]. Cigarette smoking is one of the best-described risk factors for early and premature menopause due to an antiestrogenic effect. Tobacco smoke contains many toxic polycyclic hydrocarbons that can damage ovarian germ cells and, in consequence, lead to estrogen deficiency related to follicular exhaustion [25,26]. In the case of epilepsy, in the hypothalamus, the gonadotropin-realizing hormone cell population is affected by seizures, consequently leading to anovulatory menses, lower fertility, and, finally, earlier menopause [27]. Another risk factor is being a child of multiple pregnancies. There is research that states that poor intrauterine growth, manifested as low birth weight, may lead to a decreased peak number of primordial follicles, which, in turn, may be associated with earlier menopause in adult life [25,28]. It was reported in many studies that early menarche is also a risk factor for early menopause [28]. Moreover, studies have shown positive correlations between nulliparity or low parity with premature menopause or early menopause [25,29]. Nulliparous women from around the age of 35 years who had early menarche (≤11 years) or were at risk of the occurrence of premature ovarian failure/insufficiency or early menopause and the increased risks of chronic diseases in later life were associated with earlier menopause. Due to the steroid build of estrogens, they are stored in the fatty tissue. Women with a higher body mass index (BMI) tend to have more of it than women with a lower index. A positive correlation between lower BMI and earlier or pre-menopause was reported. Women who are underweight with a BMI of 18.5 or lower are 30 percent more likely to undergo menopause before the age of 45 compared with women of a “normal” BMI between 18.5 and 22.4. This risk increases in underweight women at the age of 35. It showed a 59 percent greater likelihood of early menopause. Moreover, women aged 18 with a BMI under 17.5 were 50 percent more likely to have early menopause [25,28,30]. Interestingly, characteristics of the menstrual cycle can also be associated with early or pre-mature menopause. In the NHS II study, it was observed that short menstrual cycles and very regular cycles at the age of 18–22 years were strongly associated with a higher risk of early menopause [25,31]. It is suggested that the rate of depletion of oocytes influences the age of menopause. Women who had undergone unilateral oophorectomy were younger at menopause than women without unilateral oophorectomy. Especially high risk occurs among women with a past history of infertility associated with endometriosis [25,32].

## 5. Influence of Premature Menopause on Cognitive Functions

Naturally occurring menopause is characterized by a range of symptoms, and all these symptoms are generally referred to in a similar way in premature or early menopause. Many studies have investigated cognitive impairment in postmenopausal women, which appears to be a problem in even 70% of them. The results of a study by Woods and Mitchell have shown a noticeable decline in cognitive functions following menopause [33]. The most common complaints were problems with memory, especially difficulty in remembering words or recalling words or numbers. Betti et al. presented deficits in memory functions in Italian women during the menopausal period [34]. Based on the study by Schaafsma et al., the most common problem among menopausal women is attention deficits and reaction time, and verbal memory impairments [35]. Those conditions may result from a decreased level of estrogen. Cognitive impairments were related to stress and depression in this period and presented early (usually within the first year after the last menstrual period) [36]. 

Beyond the crucial role of estrogen in the reproductive capacity of women, there are also many other and equally important effects on the body, such as raising HDL cholesterol, reducing LDL cholesterol, causing vasodilation, and protecting against osteoporosis. The premature reduction in estrogen puts women at a higher risk for cardiovascular disease, parkinsonism, depression, osteoporosis, as well as cognitive impairment or dementia [37,38]. Moreover, premature menopause has been associated with hypertension, weight gain, and midlife diabetes, which are vascular risk factors also associated with cognitive disorders and dementia (Figure 1).

In cognitive disabilities, a crucial role is played by the estrogen receptor network, which is one of the brain’s master regulatory systems. Regions that are necessary for learning and memory, including the prefrontal cortex, hippocampus, amygdala, and posterior cingulate cortex, contain substantial estrogen receptors [47]. Estrogen helps the brain effectively respond at proper timescales to regulate brain energy metabolism, such as in the ovarian-neural estrogen axis. Changes in either the availability of estrogen or its receptor network (e.g., β-receptors) can affect intracellular signaling, neural circuit function, and energy availability in brain neurons [48]. It also has a deep impact on cortisol levels and the functional effectiveness of cortisol, the hypothalamic-pituitary-adrenal (HPA) axis, the neurotransmitters serotonin and acetylcholine, neurotrophic factors, and neuronal plasticity, as well as synaptic function. These result in stress phenomena, depression, burning sensations, and redness, as well as cognitive disorders, disturbances in verbal memory, and the worsening of the presentation of mnemonic goals [16,49,50,51]. Thus, there is a connection between cortisol, stress, and cognition; however, this hypothesis has not been extensively tested in cases of premature menopause [52]. The effect of age on cortisol activity appears to be three times higher in women than in men [53].

Thus, premature menopause affects the central nervous system (CNS) directly and indirectly, as well as both transiently and in the long term, in many complex pathways which are already discovered or yet to be.

## 6. Molecular Mechanisms Underlying the Relation between Early Menopause and Cognition

Cognitive decline rebounds as a direct consequence of the decrease in estradiol. The early and prolonged loss of ovarian-derived 17β-estradiol (E2) is the main cause of the negative neurological outcomes associated with premature menopause [54]. Experimental models have shown that estradiol prevents emotional and cognitive disorders resulting from a reduction in serotonin. Furthermore, estrogen seems to protect against cognitive disorders arising from a cholinergic deficit in women and female animals in middle age [55]. These studies suggest that the decrease in midlife estrogen levels results in changes to cholinergic and serotonergic activity, thereby contributing to emotional and cognitive disorders [55]. Estrogen may improve cerebral blood flow and increase glucose transport across the blood–brain barrier (BBB). In addition, estrogen may mediate the inhibition process of amyloid-β (Aβ) formation. Estrogen prevents Aβ from inducing a rise in the intracellular calcium level and protects against mitochondrial damage [56]. A study conducted on early postmenopausal women (around 52 years of age) showed that the effect of estrogen on serotonergic function is likely to be the main mechanism that links cognitive decline with the observed emotional disorders. The reduction in serotonin leads to emotional disorders and a large decline in verbal memory, but not spatial memory [57]. Another study employing positron emission tomography (PET) led to the conclusion that substitution treatment with transdermal estrogen led to a large increase in the binding ability of serotoninergic receptors, especially in the right frontal lobe (5-HT 2), and improved the psychomotor condition [58].

Molecular mechanisms through which E2 may regulate hippocampal memory consolidation in ovariectomized female rodents have been investigated. Frick described these mechanisms as “rapid effects of E2” on hippocampal functioning, those that occur within minutes of E2 exposure. In vivo, E2 activates numerous cell-signaling pathways. The author proposed that these effects may be mediated by non-classical actions of the intracellular estrogen receptors ERα and ERβ and by membrane-bound ERs such as the G-protein-coupled estrogen receptor (GPER) [59]. The possible effect of estrogen on hippocampal cells is presented in Figure 2. Prakapenka and Korol, in their paper, also draw attention to the fact that E2 increases extracellular levels of glucose in the hippocampus, which can lead to the hormone’s beneficial effects on hippocampus-sensitive cognition, but during that time, E2 decreases the levels of lactate and ketones in the striatum, which, in turn, can lead to the impairing effects of estradiol on striatum-sensitive cognition [60].

Moreover, menopause induced by oophorectomy in female rodents caused various influences on neurons, such as neuronal loss through apoptosis or autophagy in the prefrontal cortex, hypothalamus and amygdala, increased microglia, or astrocytes in those regions, differently expressed proteins, an adjustment in GABAergic transmission or cytoskeletal abnormalities [61]. However, these experimental models are usually created by the surgical removal of the ovaries, and these conditions are not identical to those of premature or normal menopause, which, unlike experimental models, produces luteinizing hormone (LH), follicle-stimulating hormone (FSH), gonadotropin-releasing hormone (GNRH), and testosterone [62].

Estrogen, apart from its effect on the levels and activity of neurotransmitters, promote the growth of neurons and the formation of synapses acting as antioxidants and regulating the homeostasis of calcium and the secondary messenger system [63,64,65,66,67]. The neuroprotective effects of angiotensin type 1 receptor (AT1R) blockers (ARB) in normotensive ovariectomized rats involve a reduction in plasma corticosterone and the blockade of increased AT1R activity in the hippocampus. ARBs have a therapeutic potential for normotensive women at increased risk of developing cognitive and behavioral dysfunctions due to bilateral oophorectomy prior to the natural age of menopause [68].

## 7. Mild Cognitive Impairment (MCI) in the Perimenopausal Period 

It is important to remember that mild cognitive impairment (MCI) is considered a very early stage of dementia and that dementia is very rare in people younger than 50 years of age. The risk of developing dementia increases with age, reaching 65% for individuals 65 years of age or older [69].

Most women in menopause (around 70%) report a deterioration of their memory, which includes MCI, in the peri-menopausal period [33]. Symptoms are related to the stress and depression of this period and present early (usually within the first year after the last menstrual period). Depression may be present in MCI and is difficult to ascertain whether it is depression that is causing memory dysfunctions or whether women with MCI are at a higher risk of experiencing depression. However, evidence has been presented that depression may be the first manifestation of cognitive decline [70].

In some women with MCI, dementia never occurs, and cognitive condition even improves over time. Yet, those receiving replacement therapy seem to improve their memory when the treatment is started before the last menstrual cycle [71].

## 8. Early Menopause and Vascular Implications of Dementia 

The deterioration in cardiovascular risk factors that accompany the peri-menopausal period may be another reason why cognitive functions deteriorate with age. Cardiovascular factors are also risk factors for cognitive disorders and dementia during midlife and advanced age. As a consequence, women with premature menopause and cardiovascular aggravating factors are at an even higher risk of the occurrence of mental disorders that can lead to dementia [72].

The most common forms of dementia in the elderly are Alzheimer’s disease (AD) and vascular dementia, which are two different disease entities. Nevertheless, cardiovascular diseases are pathogenetically linked to cognitive impairment and dementia regardless of the type thereof, with vascular pathology existing in at least half of patients with dementia [73]. It should be noted that the underlying pathology of dementia can begin even 20 years prior to the onset of clinical symptoms of the disease [74]. Additional studies correlating with clinical and neuropathological findings seem to support the more rapid clinical deterioration of AD when it coexists with vascular disease [75]. The well-known damage factors to the vessels and risk factors of vascular disease are hypertension, nicotine, and diabetes. All of them are characteristic of women with premature menopause in comparison with women without such a history [45]. 

Considering midlife hypertension, its impact on old age dementia has been demonstrated in studies completed in Sweden and Honolulu [53,76]. Regardless of earlier menopause, studies in Finland have shown that hypertension and hypercholesterolemia in middle-aged people themselves can result in mild MCI or AD, which will present 20 years later. This is the reason why studies have explored whether these two factors could be considered preclinical dementia markers [77].

The association of type 2 diabetes mellitus (T2DM) with increased cardiovascular disease risk (CVD) has been well documented in both genders, although women with T2DM seem to be at a higher relative CVD risk compared with their male counterparts. Women with early menopause displayed a higher risk of developing T2DM compared with women of a normal menopausal age (>45 years), including those of a late menopausal age [39]. Diabetes, if it appears before the age of 65, seems to be associated with the occurrence of mild cognitive disorders or dementia. Diabetes lasting more than 10 years has also been associated with the occurrence of mild cognitive disorders [78,79].

Another risk factor for vessel damage and corresponding cognitive impairment is increased BMI. At the age of 50, it has been associated with the increased incidence of dementia in old age, while the same event at the age of 65 shows no statistically significant relationship. Yet, we must remember that in the case of females alone, women with a lower BMI have a greater chance of early menopause, while women with high BMI usually have menopause later in life because of the estrogen stored and released from the fat tissue [46,80].

Finally, there is also ischemic leukoaraiosis which has been associated with mental disorders mainly of the frontal type (e.g., designing, using strategies, and changing objectives) [81].

## 9. Surgical Menopause and Dementia

One of the causes of earlier menopause is an oophorectomy conducted before the natural cessation of ovarian function. Indications for this surgery are multiple, including ovarian cancer, endometriosis, or tubo-ovarian abscess [82]. However, beyond the therapeutic impact of oophorectomy, there are many side effects connected to the inhibition of estrogen production due to the excision of the ovaries, e.g., cognitive impairment [83]. Mental consequences depend, however, on the time of the surgery. The Mayo Clinic Cohort Study of Oophorectomy and Aging proved that the risk of cognitive impairment and dementia increase in women who have an oophorectomy at a younger age. When a unilateral oophorectomy was performed before 41 years of age, there was an almost doubled risk, and when it was performed before 34 years of age, this risk was more than four times higher. The study shows that the increased risk among these women was restricted to those who did not take estrogen after surgery until at least the age of 50 years. Thus, it is suggested that estrogen treatment following oophorectomy protects against the increased risk of cognitive impairment or dementia. A higher risk of parkinsonism and connected dementia, which increased in younger patients undergoing oophorectomy, was also observed. However, no associations were reported between estrogen use and cognition in women with surgical or natural menopause, which took place in older age [84,85,86,87,88,89].

In other studies, women who underwent surgical menopause also showed significantly decreased global cognitive functioning scores and memory scale scores 3 to 6 months after an oophorectomy compared with premenopausal controls [84]. Yet, hormone therapy may ameliorate symptoms of dementia when taken within the first 6 months after the surgery [90,91,92]. It was also observed that, even though the global brain structure after hormonal therapy discontinuation decreased, the white matter hyperintensities tended to continually increase [93]. However, estradiol in hormone therapy could help maintain working memory only without improving verbal memory [90,91,92]. Meta-analysis on this topic confirmed this evidence and underlined that the younger the patient undergoing oophorectomy was, the more rapid the decline in cognitive function [94].

Surgical premature menopause also increases the risk of AD. The beginning of the changes in the biochemistry of the brain connected with AD converges with the onset of menopause. It is widely known that changes begin decades prior to the occurrence of clinical symptoms. Although women tend to live longer than men, this factor is not considered the only reason why females are about two-thirds of people living with AD [95]. Work on an animal model of premature menopause (long-term ovariectomy) revealed that the hippocampus, an area critical in learning and memory, becomes hypersensitive to ischemic injury following surgical menopause. Further work showed that the brain’s hypersensitivity also extended to an AD-relevant insult, as the hippocampus of surgically menopausal rats was profoundly hypersensitive to the neurotoxic effects of Aβ. The hippocampal hypersensitivity and Aβ may also induct the NADPH oxidase/superoxide/C-Jun N-terminal kinase pathway: a stress-activated intracellular signaling cascade that further enhances oxidative stress and promotes apoptosis in hippocampal neurons through both transcriptional and post-translational mechanisms [96].

The research on the female triple transgenic mouse model of AD indicated that reproductive senescence in the normal non-transgenic brain parallels the shift to the ketogenic/fatty acid substrate phenotype with a concomitant decline in mitochondrial function and exacerbation of bioenergetic deficits in the transgenic brain. In triple-transgenic Alzheimer’s mice, ovariectomy significantly exacerbated mitochondrial dysfunction and induced mitochondrial Aβ and Aβ-binding-alcohol-dehydrogenase (ABAD) expression [97]. It is essential to point out that hypometabolism, reduced mitochondrial function, and subsequent oxidative damage are known to promote the accumulation of Aβ pathology and neuronal dysfunction, therefore increasing the risk of developing AD later in life [98].

Another explanation of the impact of early menopause associated with oophorectomy on dementia is the abrupt cessation of exposure to ovarian hormones, which accelerates cognitive aging. There is also an epigenetic signature of aging shown in DNA methylation data from human blood, saliva, and buccal epithelium, suggesting an earlier menopausal transition [99]. This can be supported by the epidemiological data showing a higher all-cause mortality among premenopausal women who underwent the removal of ovaries to those conserving ovaries at the time of the hysterectomy for benign disease [100].

Hormonal alterations that occur after the surgical removal of the ovaries, natural menopause, or the administration of exogenous estrogens to menopausal women may also lead to an altered immune profile and, in turn, to changes in cognitive processes [101]. Generally, estradiol can reduce the Aβ level and tau hyperphosphorylation as well as the inflammatory cascade [102]. However, estrogen decline during the menopausal transition is associated with the presence of chronic low-grade inflammation - both in the periphery and in the brain. This state is associated with increased peripheral pro-inflammatory cytokines, e.g., IL-1, IL-6, IL-8, or TNF-α via the induction of the transcription factor NFĸB and alteration in the cellular immune response, which leads to a pro-inflammatory microenvironment in the brain (neuroinflammation) [103]. Evidence showed that the upregulation of inflammatory genes in the brain is not solely related to age but may be a result of E2 when decreased in menopause [101]. Moreover, low levels of E2 may lead to increased levels of TNF-α and the stimulation of Th17 T-cells, causing increased IL-17, which could play a primary role in chronic inflammation. Thus, inflammation may mediate the relationship between systemic E2 decline, cognitive changes, and dementia (Figure 3). 

## 10. Early and Premature Menopause and Hormone Replacement Therapy (HRT)

Menopausal hormone therapy from menopause to the age of 60 years may provide a “window of opportunity” to reduce the risk of MCI and AD in later life. However, some studies have shown conflicting results, making the impact of estrogen exposure on dementia risk even more complex. For example, a 44-year longitudinal population study on Swedish women by Najar et al. concluded that a longer reproductive period was associated with an increased risk of dementia due to the toxic effects of endogenous estrogen in late life [104]. There is also evidence that hormone therapy may have potentially adverse effects on cognition, mostly in women who have health risks such as lower global cognition or diabetes [13] or has no effect at all [105,106]. Both premature surgical menopause and premature ovarian failure are associated with long-term negative effects on cognitive function, which are not entirely offset by menopausal hormone therapy. In terms of surgical menopause, potential long-term effects on cognitive function should form part of the risk/benefit ratio when considering ovariectomy in younger women, especially carriers of the *APOEε4* allele [107,108]. An alternative prevention and therapy method for cognitive impairment seems to be physical activity, which in middle age leads to great cognitive gain [109].

Concerning vessel diseases and cardiovascular morbidity and mortality, most studies have shown the beneficial effect of hormonal therapy in women at early menopausal age (<10 years since the final menstrual period) or younger than 60 years. Hormonal therapy is recommended in women with early menopause and POI. It has been shown that hormonal therapy when taken early after menopause, can prevent the reduction in prefrontal cortex activity: a measure observed in early dementia [110]. What is more, transdermal estrogens have a lower risk of thrombosis compared with oral regimens [82].

These findings only underline the need for the prudent implementation of hormone replacement therapy (HRT) and the necessity of conducting further prospective studies. Many measures may help to prevent, delay, or minimize AD in both women and men and should be actively encouraged [111]. The impact of estrogen therapy in early menopause and prolonged reproductive period on cognitive impairment is presented in Figure 4.

## 11. Conclusions

Premature and early menopause affects the central nervous system directly and indirectly, both transiently and in the long term, leading to cognitive impairment or even dementia, mainly due to a decrease in estrogen levels and comorbidity with cardiovascular risk factors, autoimmune diseases, and aging. The women affected by premature or early menopause are also at higher risk for other diseases, e.g., AD. It is well known that women are nearly twice as often affected by AD compared to men. Until now, the reason for this imbalance has been unclear. The disproportionate impact of Alzheimer’s on women is attributed to age: women live longer than men, and age is the greatest known risk factor for Alzheimer’s, but probably not the only one. Researchers are looking at a variety of factors, including those that are biological and those that are social, and cultural. One area of research is the female reproductive period. There are associations between the risk of dementia and the age of menarche, the age of menopause, and the time between first menstruation and menopause. Another interesting problem regards the sex-specific differences in the architecture of the brain. Differences found in the structural and functional connections of a woman’s brain may speed the spread of tau: a protein that clumps into tangles and may contribute to cell damage and, ultimately, cell death. This finding could lead to the creation of risk-reduction strategies targeted at women.

This review emphasizes the need for the early identification of women with earlier menopause as a group that is at a higher risk of cognitive decline. These women should be taken care of by various specialists such as gynecologists, endocrinologists, neurologists, and psychiatrists in order to maintain their mental health to the highest possible level. It is also important to consider the impact of surgical menopause on the patient during the treatment of other diseases, such as whether it is better to postpone the surgical procedure and try different types of treatment until the age of natural menopause. When a woman has undergone an oophorectomy, she should be offered complex care, which should also take into account her future cognitive abilities. 

Finally, the correlation between earlier menopause and dementia still needs to be fully discovered as long as it includes an abundance of possible pathways. Analyzing them in the context of the clinical diversity of patients would significantly improve the prevention, diagnostics, and therapy of dementia among women with earlier menopause. 

## Figures and Tables

**Figure 1 ijms-24-06566-f001:**
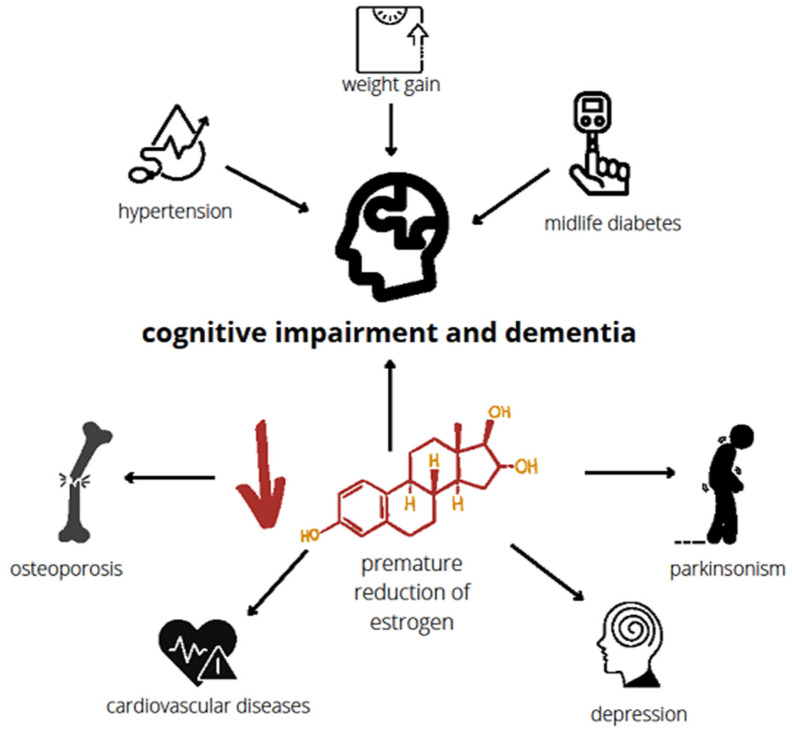
Risks following premature reduction in estrogen. The characteristic of premature and early menopause is a diminished level of estrogen that occurs before the physiological time of menopause. This reduction in estrogen increases the risk of dementia as well as cardiovascular disorders, osteoporosis, depression, and parkinsonism. Cognitive disorders and dementia could also be induced by hypertension, weight gain, and midlife diabetes which develop frequently in individuals with menopause [39,40,41,42,43,44,45,46].

**Figure 2 ijms-24-06566-f002:**
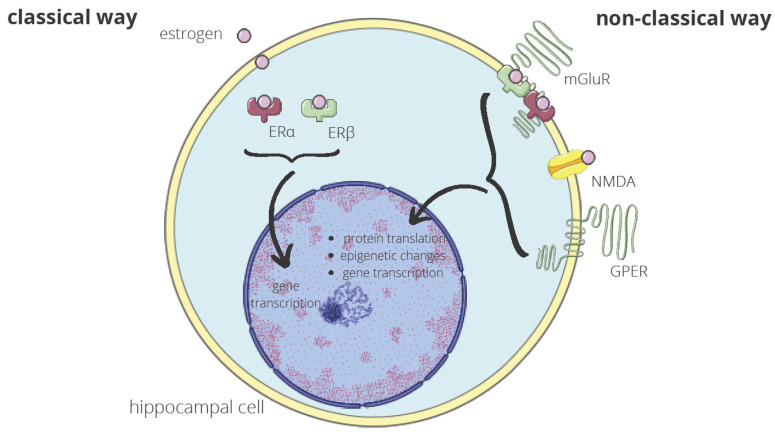
Effect of estrogen on hippocampal cells. Estrogen interacts with the hippocampal cells through a classical and non-classical way. In the classical one, estrogen binds to ERα and ERβ in the cytoplasm. The estrogen-receptor complexes interact then with the genome, inducing gene transcription. The non-classical way takes place near or in the plasma membrane. Metabotropic glutamate receptor 1a (mGluR1a) is influenced by ERα and ERβ with the bound estrogen, the N-methyl-D-aspartate receptor (NMDA), by estrogen alone and GPER by estrogen indirectly. The receptors induce cell signaling, resulting in protein translation, epigenetic changes, or gene expression. These molecular processes are crucial for memory formation in the hippocampus [59].

**Figure 3 ijms-24-06566-f003:**
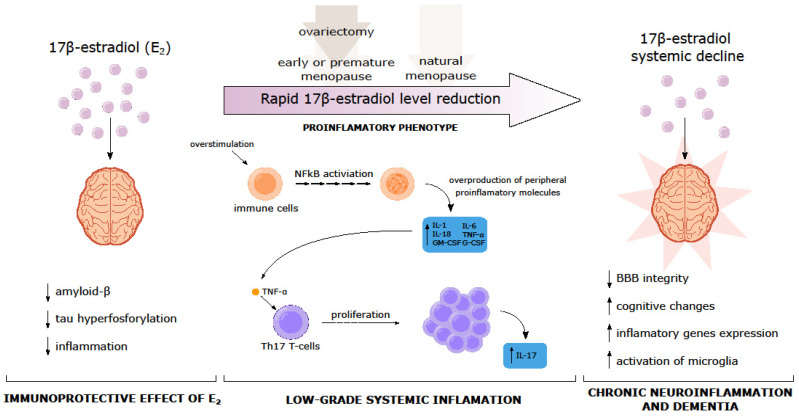
Relationship between systemic E2 decline, chronic inflammation, cognitive changes, and dementia. Relationships exist between sex hormones, predominantly estrogen and cognition. The key molecular mechanisms underlying cognitive decline during the menopausal transition are presented, including those with both inflammatory pathways (e.g., activation of pro-inflammatory genes, activation of microglia) and inflammatory components (over-production of inflammatory molecules). It is proposed that rapid estrogen decline in early or premature as well as natural menopause and is associated with the presence of chronic low-grade inflammation, both in the periphery and in the brain.

**Figure 4 ijms-24-06566-f004:**
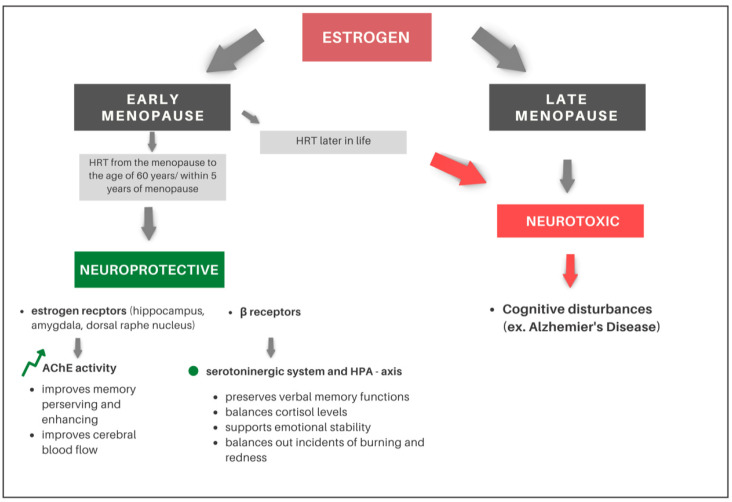
The impact of estrogen therapy in early menopause and prolonged reproductive period on cognitive impairment. At the present stage of knowledge about the pathomechanism of cognitive decline in postmenopausal women, it can be assumed that estrogen can play neuroprotective or neurotoxic role sin early menopause and late menopause, respectively. Women who experience menopause later in life have higher estrogen levels when they are more advanced in age and that is the time when dementia-related pathologies accumulate. Therefore, higher estrogen levels may become neurotoxic in older age, which explains why HRT turns out to be beneficial in terms of dementia risk if administrated within 5 years of the menopause but has negative effects if administrated after that time.

**Table 1 ijms-24-06566-t001:** Causes of premature menopause.

Etiology	Effects	Ref.
**Inheritance**	Both numerical and structural changes in the X chromosome are the most common imbalances affecting about 13% of the POI cases	[15,16]
**Autoimmune diseases**	Antibodies produced in these diseases may also attack the ovaries: Hashimoto’s thyroiditis Diabetes rheumatoid arthritis	[17]
**Primary ovarian insufficiency (POI)**	The inhibition of the ovaries’ function before the age of 40 remains in close relationship with premature menopause. The causes of POI include the following: iatrogenic causes (ovarian surgery, radiotherapy or chemotherapy) environmental factors viral infections metabolic diseases autoimmune diseases genetic alterations.	[18]

**Table 2 ijms-24-06566-t002:** Causes of early menopause.

Etiology	Effects	Ref.
**Genetic abnormality **	Turner syndrome (one of the X chromosomes is missing or abnormal) Fragile X syndrome (where the bottom of the long arm of the X chromosome is broken or fragile) Women who have Turner’s syndrome type XO and those who are carriers of fragile X often have POI	[19,20]
**Autoimmune disorders **	thyroid disease Type 1 diabetes Crohn’s disease coeliac disease chronic candidiasis (thrush)	[17,21]
**Metabolic disorders **	galactosaemia aromatase deficiency (a problem in converting the hormone androgen to estrogen)	[20,22]
**Infection **	mumps oophoritis pelvic tuberculosis varicella Shigella	[20]
**Idiopathic **	Individual cases of women whose periods stop with no known cause	[23]

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
