# Peer review of "Cognitive Decline in Early and Premature Menopause"

_ijms, 2023, doi:10.3390/ijms24076566_

Round 1

Reviewer 1 Report

I enjoyed reading the review by Sochocka et al. on cognitive decline in early and premature menopause. The paper is well written and comprehensive; however, it would benefit from some English editing. It brings into light an important topic. An emerging area of research is the influence of the gut microbiome on menopause and potential impact of the gut microbiome on menopause-related disease risk. While it can be a separate review, it may be beneficial to the readers if the authors would provide information on the role of the gut microbiome in menopause and cognitive decline. This would also enhance the novelty of this review.

Author Response

Reviewer 1

I enjoyed reading the review by Sochocka et al. on cognitive decline in early and premature menopause. The paper is well written and comprehensive; however, it would benefit from some English editing.

Response: Thank you for this suggestion. English language has been improved.

It brings into light an important topic. An emerging area of research is the influence of the gut microbiome on menopause and potential impact of the gut microbiome on menopause-related disease risk. While it can be a separate review, it may be beneficial to the readers if the authors would provide information on the role of the gut microbiome in menopause and cognitive decline. This would also enhance the novelty of this review.

Response: This is an excellent advice! Thank you! Although we did not focus on this topic it was valuable to add this information to the main text. I think it improved the manuscript.

Reviewer 2 Report

The authors present an interesting review on effects of early and premature menopause on cognitive decline. This paper is well-tructured and might be a nice review article. Nevertheless, it requires some revision, especially related to the presentation. Specific points are listed below.

1. English usage is very uneven throughout the manuscirpt. Some parts are well written while others require substantial revision of grammar, style and interpunction. Thus, it is recommended to check the whole text in these aspects.

2. Abstract can be better organized. Please, try to present the major problem(s) first, then indicate possible mechanism(s), and finally describe possible treatment options.

3. Line 45: What do the authors mean by saying "in our environment"? Please, be more precise.

4. The description of methods should not be included as the last paragraph of Intruction, but it should be presented as a separate chapter. Then, the methods should be described more precisely. How many records were obtained after the described search options? What was the further procedure to select papers discussed and cited in this manuscript. How many articles fulfilled the additional requrements?

5. Tables 1 and 2 are confusing. They do not contain titles of columns, thus, it is unclear what was the purpose of the presentation. The descriptions are very long, making the tables hard to follow. Please, either modify the tables to indicate specific points you want to demonstrate by naming columns and providing very short information in each table cell, or delete the tables and describe relevant fatcs in the main text.

6. Lines 85-111: This fragment contains some bullets indicating the risk factors and their short descriptions. However, this fragment of the manuscript looks like a part of dissertation rather than like a scientific article. Please, consider either preparation of an informative table (not like Tables 1 and 2) or description of these risk factors in the form of a regular text (the latter option is prefered).

7. Lines 112 and 158: Chapters starting at these lines have the same numbers (4). Please, correct the numbering of chapters.

8. Lines 119, 169, and others: Starting a sentence with "Also" is grammatically correct, however it makes the sentences awkward. Please, re-write.

9. Figure 1: Why did the authors suggest that hypertension, weight gain, and midlife diabetes are dependent on cognitive impairment and dementia, as shown in the figure? Alternatively, did I misunderstand the picture? If the latter is true, the figure is confusing and should be corrected. Perhaps the use of arrows instead of lines would improve the clarity and indicate the direction of proposed relationships (what is the cause and what is the effect).

10. Line 138: What "these last two pathologies"? It is not clear what pathologies are the authors talking about.

11. Line 150: The authors wrote "We could make the claim....", but then they cited an article published by others (ref. 40). So, who was the author of this claim?

12. Line 165: Provide relevant reference(s) to these statements.

13. Lines 180-181: How can "rapid effects of E2 on hippocampal cell signaling" be "necessary for E2 to facilitate the consolidation". The logic of this sentence is quite strange. Please, re-write and express your idea more precisely.

14. Lines 259-260: This sentence is awkward and unclear. Please, re-write.

15. Please, use one name for specific items throughout the text. For example, page 8 contains three different names for the same molecule: "amyloid-beta", "beta (Greek letter) amyloid", and "A beta (Greek letter)". This makes the text confusing.

16. Please, use appropriate genetic and biochemical nomenclature throughout the text. For example, names of human genes should be italicized.   

17. Line 386: Did the authors mean "respectively" instead of "retrospectively"? If so, please, correct.

18. Line 397: What did the authors mean by saying "this study"? This is a review article, not an original paper. Please, clarify.

19. Line 403: What did the authors mean by saying "the woman"? What particular woman is described? Did you mean "a woman"?  

Author Response

Reviewer 2

The authors present an interesting review on effects of early and premature menopause on cognitive decline. This paper is well-structured and might be a nice review article. Nevertheless, it requires some revision, especially related to the presentation. Specific points are listed below.

  1. English usage is very uneven throughout the manuscript. Some parts are well written while others require substantial revision of grammar, style and interpunction. Thus, it is recommended to check the whole text in these aspects.

Response: All manuscript has been corrected by Native Speaker to improve the language.

  1. Abstract can be better organized. Please, try to present the major problem(s) first, then indicate possible mechanism(s), and finally describe possible treatment options.

Response: Thank you for this valuable suggestion. The Abstract has been changed.

  1. Line 45: What do the authors mean by saying "in our environment"? Please, be more precise.

Response: Thank you for pointing on this confusing sentence. It has been changed.

  1. The description of methods should not be included as the last paragraph of Introduction, but it should be presented as a separate chapter. Then, the methods should be described more precisely. How many records were obtained after the described search options? What was the further procedure to select papers discussed and cited in this manuscript. How many articles fulfilled the additional requirements?

Response: Thank you for this suggestion. This is a narrative review. We focused to obtaining a broad perspective on topic of the associations between early and premature menopause with cognitive disorders. We specified a research hypothesis and search as much newest publication on this topic but not according to systematic review criteria. The efficacy of narrative reviews is irreplaceable in tracking the development of a scientific principle, or a clinical concept. This ability to conduct a wider exploration could be lost in the restrictive framework of a systematic review. We aimed at identifying and summarizing what has previously been published. However we agree that this section should be more precisely presented. We have re-write this part.

  1. Tables 1 and 2 are confusing. They do not contain titles of columns, thus, it is unclear what was the purpose of the presentation. The descriptions are very long, making the tables hard to follow. Please, either modify the tables to indicate specific points you want to demonstrate by naming columns and providing very short information in each table cell, or delete the tables and describe relevant facts in the main text.

Response: We absolutely agree with this comment. Tables have been carefully checked and changed according to your suggestions.

  1. Lines 85-111: This fragment contains some bullets indicating the risk factors and their short descriptions. However, this fragment of the manuscript looks like a part of dissertation rather than like a scientific article. Please, consider either preparation of an informative table (not like Tables 1 and 2) or description of these risk factors in the form of a regular text (the latter option is preferred).

Response: Thank you for this valuable suggestion. We have re-write this part and prepared as a regular text.

  1. Lines 112 and 158: Chapters starting at these lines have the same numbers (4). Please, correct the numbering of chapters.

Response: Thank you for pointing on this mistake. It has been changed.

  1. Lines 119, 169, and others: Starting a sentence with "Also" is grammatically correct, however it makes the sentences awkward. Please, re-write.

Response: Thank you for this suggestion. All have been re-write.

  1. Figure 1: Why did the authors suggest that hypertension, weight gain, and midlife diabetes are dependent on cognitive impairment and dementia, as shown in the figure? Alternatively, did I misunderstand the picture? If the latter is true, the figure is confusing and should be corrected. Perhaps the use of arrows instead of lines would improve the clarity and indicate the direction of proposed relationships (what is the cause and what is the effect).

Response: We absolutely agree with this comment. Figure has been changed to avoid any uncertainties.

  1. Line 138: What "these last two pathologies"? It is not clear what pathologies are the authors talking about.

Response: Thank you for pointing on this confusing sentence. It has been changed.

  1. Line 150: The authors wrote "We could make the claim....", but then they cited an article published by others (ref. 40). So, who was the author of this claim?

Response: Thank you for pointing on this confusing sentence. It has been changed.

  1. Line 165: Provide relevant reference(s) to these statements.

Response: Thank you for this suggestion. We have added citation.

  1. Lines 180-181: How can "rapid effects of E2 on hippocampal cell signaling" be "necessary for E2 to facilitate the consolidation". The logic of this sentence is quite strange. Please, re-write and express your idea more precisely.

Response: Thank you for pointing on this confusing part. It has been changed.

  1. Lines 259-260: This sentence is awkward and unclear. Please, re-write.

Response: Thank you for pointing on this confusing sentence. It has been changed.

  1. Please, use one name for specific items throughout the text. For example, page 8 contains three different names for the same molecule: "amyloid-beta", "beta (Greek letter) amyloid", and "A beta (Greek letter)". This makes the text confusing.

Response: Thank you for pointing on this mistake. It has been changed.

  1. Please, use appropriate genetic and biochemical nomenclature throughout the text. For example, names of human genes should be italicized. 

Response: Thank you for pointing on this mistake. It has been changed.

  1. Line 386: Did the authors mean "respectively" instead of "retrospectively"? If so, please, correct.

Response: Thank you for pointing on this mistake. The sentence has been changed.

  1. Line 397: What did the authors mean by saying "this study"? This is a review article, not an original paper. Please, clarify.

Response: Thank you for pointing on this mistake. The sentence has been changed.

  1. Line 403: What did the authors mean by saying "the woman"? What particular woman is described? Did you mean "a woman"?  

Response: Thank you for pointing on this mistake. The sentence has been changed.

Reviewer 3 Report

Introduction Method: 

In the literature search, it is necessary to describe exclusion conditions

How to present Figures and Tables :  Inappropriate      

Table 1  Table 2    :  I don't understand the meaning of bold      

Figure 1  Figure 2 :  The text and the description of the figure are confused 

Summary:

It is well known that there are gender differences in the manifestation of dementia.

It is insufficient as a review that the comparison with men is not described.

Author Response

Reviewer 3

  1. Introduction Method:

In the literature search, it is necessary to describe exclusion conditions

Response: Thank you for this suggestion. This is a narrative review. We focused to obtaining a broad perspective on topic of the associations between early and premature menopause with cognitive disorders. We specified a research hypothesis and search as much newest publication on this topic but not according to systematic review criteria. The efficacy of narrative reviews is irreplaceable in tracking the development of a scientific principle, or a clinical concept. This ability to conduct a wider exploration could be lost in the restrictive framework of a systematic review. We aimed at identifying and summarizing what has previously been published. However, we agree that this section should be more precisely presented. We have re-write this part.

  1. How to present Figures and Tables :  Inappropriate      

Table 1  Table 2    :  I don't understand the meaning of bold      

Figure 1  Figure 2 :  The text and the description of the figure are confused 

Response: We absolutely agree with this comment. Tables, Figures and Figures’ legends have been changed to avoid any uncertainties.

  1. Summary:

It is well known that there are gender differences in the manifestation of dementia. It is insufficient as a review that the comparison with men is not described.

Response: Thank you for this valuable suggestion. We have added an additional information regarding gender differences in the manifestation of dementia.

Round 2

Reviewer 3 Report

I think this is an advanced and valuable research.

Please reconsider the layout, such as the division of Table 2 and the blank space on page 11.